# Peroxisomal Localization of a Truncated HMG-CoA Reductase under Low Cholesterol Conditions

**DOI:** 10.3390/biom14020244

**Published:** 2024-02-19

**Authors:** Jianqiu Wang, Markus Kunze, Andrea Villoria-González, Isabelle Weinhofer, Johannes Berger

**Affiliations:** Department of Pathobiology of the Nervous System, Center for Brain Research, Medical University of Vienna, 1090 Vienna, Austria

**Keywords:** HMGCR, statin, lovastatin, peroxisome, dual localization, PTS2, organelles

## Abstract

3-hydroxy-3-methylglutaryl-CoA reductase (HMG-CoA reductase, HMGCR) is one of the rate-limiting enzymes in the mevalonate pathway required for cholesterol biosynthesis. It is an integral membrane protein of the endoplasmic reticulum (ER) but has occasionally been described in peroxisomes. By co-immunofluorescence microscopy using different HMGCR antibodies, we present evidence for a dual localization of HMGCR in the ER and peroxisomes in differentiated human monocytic THP-1 cells, primary human monocyte-derived macrophages and human primary skin fibroblasts under conditions of low cholesterol and statin treatment. Using density gradient centrifugation and Western blot analysis, we observed a truncated HMGCR variant of 76 kDa in the peroxisomal fractions, while a full-length HMGCR of 96 kDa was contained in fractions of the ER. In contrast to primary human control fibroblasts, peroxisomal HMGCR was not found in fibroblasts from patients suffering from type-1 rhizomelic chondrodysplasia punctata, who lack functional PEX7 and, thus, cannot import peroxisomal matrix proteins harboring a type-2 peroxisomal targeting signal (PTS2). Moreover, in the N–terminal region of the soluble 76 kDa C-terminal catalytic domain, we identified a PTS2-like motif, which was functional in a reporter context. We propose that under sterol-depleted conditions, part of the soluble HMGCR domain, which is released from the ER by proteolytic processing for further turnover, remains sufficiently long in the cytosol for peroxisomal import via a PTS2/PEX7-dependent mechanism. Altogether, our findings describe a dual localization of HMGCR under combined lipid depletion and statin treatment, adding another puzzle piece to the complex regulation of HMGCR.

## 1. Introduction

The enzyme 3-hydroxy-3-methylglutaryl-coenzyme A reductase (HMGCR, NCBI Reference Sequence: NP_000850.1) exerts the rate-limiting step of the mevalonate pathway required for the biosynthesis of cholesterol [1] and essential non-sterol metabolites such as farnesyl pyrophosphate (FPP), geranylgeranyl pyrophosphate (GGPP), dolichol, heme A, isopentenyl adenosine and ubiquinone (coenzyme Q) [2,3]. Reduced activity of HMGCR caused by biallelic variants can cause autosomal recessive progressive limb-girdle muscular dystrophy [4]. HMGCR is a highly regulated enzyme [5] affected by a complicated multilevel feedback system, which acts on enzyme activity, protein stability, gene expression and translation efficiency [6,7,8,9]. Thus, competitive HMGCR inhibitors such as statin drugs (e.g., lovastatin) [10,11], which efficiently reduce the mevalonate level [12], and thus the pools of sterols and other isoprenoids, also result in a drastic upregulation of HMGCR [13]. Human HMGCR is an integral membrane protein of the endoplasmic reticulum (ER) [14,15] consisting of 888 amino acids (AAs) [16] encompassing a highly conserved transmembrane-8 span domain of 330 AAs [17,18]; a long, flexible linker region of 120 AAs; and a widely conserved C-terminal catalytic domain of 430 AAs [19]. In the presence of excess sterols, HMGCR becomes susceptible to proteolysis and can be effectively degraded by an ER-associated degradation (ERAD) [20,21] process, which is induced by a combined increase in cellular sterol and non-sterol isoprenoids [22,23] and mediated by insulin-induced gene (INSIG) protein-activated ubiquitination and a proteasome-dependent degradation pathway [24,25,26,27,28]. This degradation involves several proteolytic processing steps, and several cleavage sites have been confirmed within the transmembrane domain and the linker region [16]. It has been shown that the sterol-induced dislocation of HMGCR from the ER membrane into the cytosol involves a subcellular compartment resembling lipid droplets [29]. In rodents, an enzymatically active C-terminal fragment of 62 kDa is released from the ER membrane and can be further processed to a 53 kDa fragment without losing enzymatic activity [16]. 

In 1985, Keller and colleagues described HMGCR in peroxisomes by using immuno-gold labeling [30], and these results were later reconfirmed [31,32]. Peroxisomes are ubiquitous, single-membrane-bounded organelles, which were first characterized to have important functions in lipid metabolism and oxidative stress [33,34]. Their importance is underscored by the existence of a wide range of inherited human diseases linked to a complete or partial dysfunction of peroxisomes [35,36,37,38,39]. Next to their indispensable role in multiple metabolic pathways such as fatty acid β-oxidation [40] or plasmalogen biosynthesis [41,42], peroxisomes contribute to various other cellular processes, including signaling [43], viral response and antiviral immunity [44,45]. 

Many publications have reported the peroxisomal location of enzymes involved in cholesterol biosynthesis [46,47,48,49,50,51,52]. However, the claim of a peroxisomal contribution to cholesterol biosynthesis [53] has been disputed due to results contradicting the concept of a peroxisomal segment of cholesterol biosynthesis [54,55]. In particular, the peroxisomal compartmentation of mevalonate kinase, phosphomevalonate kinase and mevalonate pyrophosphate decarboxylase has been questioned [56,57,58]. However, various other cellular and physiological findings corroborate a link between peroxisomes and cholesterol. In the absence of peroxisomes, the cholesterol level, the gene expression of enzymes involved in cholesterol biosynthesis [51,59,60,61,62] and cholesterol trafficking [63,64] are changed. Moreover, acetyl-CoA produced inside peroxisomes from the degradation of very-long-chain fatty acids (VLCFAs) was effectively integrated into cholesterol [65]. Defects in ABCD1, the peroxisomal transporter for VLCFAs, are accompanied in human cells by dysregulated cholesterol homeostasis and transport [66], and in *Abcd1*-deficient mice cholesterol levels are increased [67]. Conversely, the treatment of keratinocytes with HMGCR inhibitors increased peroxisome number and activity [68]. All these results suggest the embedding of peroxisomes within a net of cellular processes jointly ensuring cholesterol homeostasis, which includes the synthesis, uptake, export, esterification and hydrolysis of cholesterol esters, but also the intracellular storage of cholesterol in lipid droplets. The physical interactions between peroxisomes and other membrane-bound compartments by means of a tethering mechanism, which is mediated by local contact sites, allows an effective transfer of metabolites [69]. For cholesterol and other isoprenoids, the interaction with the ER [70,71] is indispensable, but also the interaction of peroxisomes with lipid droplets [72] and lysosomes [63] may play important roles in cholesterol homeostasis [73]. 

In this study, we provide evidence for a bi-localized distribution of HMGCR between ER and peroxisomes and for a PTS2/PEX7-mediated peroxisomal targeting of the soluble, catalytic domain of HMGCR under cholesterol-depleted conditions. 

## 2. Materials and Methods

### 2.1. Cell Culture

Cell lines: The mouse A9 Hybridoma cell line, which was used to produce the HMGCR-A9 antibody, was obtained from the American Type Culture Collection (ATCC, CRL-1811). The HEK-293 cell line was also obtained from ATCC (CRL-1573). HeLa cells were derived from the European Collection of Authenticated Cell Culture (ECACC -86090201). The human hepatoma cell line HepG2 was also obtained from ATCC (CRL-11997). The human monocytic cell lines THP-1 and U-937 were obtained from ATCC (TIB-202 and CRL-1593.2). The human fetal microglial cell line CHME3 was a gift from Dr. Karsten Tedin (Department of Microbiology and Genetics, University of Vienna). Human primary skin fibroblasts from RCDP1 and RCDP2 patients with mutations in the *PEX7* or *GNPAT* gene were previously described [74,75] and were provided by Nancy E. Braverman (Department of Human Genetics and Pediatrics, McGill University, Montreal, Canada). Informed consent for the use of the patient cell lines for research was obtained by the McGill University Health Center according to institutional guidelines. Control fibroblasts derived from metabolically healthy individuals were obtained from Dr. Brunhilde Molzer (Institute for Neuropathology, Medical University of Vienna). All studies involving human fibroblasts were approved by the Ethical Review Board of the Medical University of Vienna (application no. EK729/2010). Primary human CD14+ monocytes from healthy donors were isolated from leukocyte reduction chambers, purchased from the General Hospital of Vienna, using Ficoll density gradient centrifugation (PAN Biotech, Aidenbach, Germany) and positive selection for CD14+ cells using MACS microbeads and an LS column system (Miltenyi Biotec, Bergisch Gladbach, Germany) according to the manufacturers’ instructions, as recently described [76]. All studies involving human monocytes/macrophages were approved by the Ethical Review Board of the Medical University of Vienna (application no. EK1462/2014).

Culture medium: A9 Hybridoma, CHME3, HEK-293, HeLa, HepG2 and U-937 cells were cultivated in Dulbecco’s modified Eagle’s medium (DMEM, PAA), supplemented with 10% (*v*/*v*) heat-inactivated fetal bovine serum (FBS, PAA), 2 mM l-glutamine (Lonza, Basel, Switzerland), 100 units/mL penicillin (Lonza), 100 μg/mL streptomycin (Lonza) and 1 μg/mL Fungizone (Invitrogen, Carlsbad, CA, USA) as a complete medium in an atmosphere of 5% CO_2_ at 37 °C. Human fibroblasts, monocytes and THP-1 cells were cultivated in Roswell Park Memorial Institute medium (RPMI-1640, PAA) with the same supplementation.

Cell stimulation and differentiation: CHME3 cells were activated in DMEM for 24 h using 50 ng/mL IFNγ (Immuno Tools). Primary human CD14+ monocytes were differentiated to macrophages using RPMI-1640 supplement with 10% LPS-free FCS (Gibco Life Technologies/Invitrogen, Waltham, MA, USA) and 50 ng/mL M-CSF (PeproTech, Rocky Hill, NJ, USA) for 7 days. U-937 cells were differentiated in DMEM using 40 ng/mL phorbol 12-myristate 13-acetate (PMA, Sigma, St. Louis, MA, USA) for 3 days. THP-1 cells were differentiated in RPMI using 40 ng/mL PMA for 3–7 days.

Sterol depletion: Lipid-depleted medium (LDM) was supplemented with lipid-depleted FBS (Bodinco, Alkmaar, The Netherlands, BDC-5608) as a replacement for FBS for 3 or 7 days, and with 5 μM lovastatin overnight (for the last 16 h). For primary human in vitro-differentiated macrophages, only 8 h of lovastatin incubation was carried out due to the sensitivity of this primary cell type. See also Table 1 for detailed treatment conditions. 

### 2.2. Preparation of Cells for Immunofluorescence Microscopy

Cells were seeded onto 24 × 24 mm glass coverslips and incubated under the indicated growth conditions until they were fixed for 15 min using 3.7% paraformaldehyde in phosphate-buffered saline (PBS), followed by 0.1% Triton X-100 for permeabilization for 5 min, and then blocked in blocking solution (PBS with 10% FCS and 5% bovine serum albumin, Roche Applied Science, Penzberg, Germany) for 2 h at room temperature. Cells were incubated with primary antibodies from different species overnight at 4 °C under a humidified atmosphere. Immunostained coverslips were then incubated with the corresponding secondary antibody for 1 h at room temperature, counterstained by DAPI (4′,6-diamidino-2-phenylindole, Roche Diagnostics Gmbh, Mannheim, Germany) to visualize nuclei, and mounted on glass slides with Mowiol (Sigma, St. Louis, MI, USA)-based mounting medium. Coverslips were rinsed in PBS between each step. Slides were later analyzed with an Olympus invert microscope IX71 equipped with a CCD camera (CAM-XM10) using Cell^M/Cell^R software (version 3.2) (Olympus, Shinjuku, Japan) or a confocal laser scan microscope (Leica SP5, Leica, Mannheim, Germany) using Leica confocal LAS AF software (version 3.3.10134). During the analysis of subcellular distribution, cells showing apoptosis or extremely high expression levels were avoided.

### 2.3. Antibodies for Immunofluorescence Microscopy and Western Blot Analysis

The primary antibodies used in the experiments were as follows: mouse α-ABCD1 (Euromedex, Souffelweyersheim, France); mouse α-β-Actin (Chemicon, Tokyo, Japan); mouse α-ATP synthase (Molecular Probes, Eugene, OR, USA); a rabbit polyclonal α-calnexin antibody, which was a gift from Erwin Ivessa (Max Perutz Labs, Vienna Biocenter, Vienna, Austria) and is directed against the COOH-terminal peptide of calnexin (AA 555–573 of the mature dog protein [77]; mouse α-GRP78 (BD Transduction Laboratories, Franklin Lakes, NJ, USA); mouse α-HMGCR (antibody was prepared from supernatant of A9 hybridoma cells, ATCC #CRL-1811 directed against HMGCR AAs 621-825); mouse α-PDI (Stressgen, Victoria, BC, Canada); rabbit α-PMP70 (Genetex, Irvine, CA, USA); and rabbit α-HMGCR (a gift from Prof. Peter A. Edwards, Department of Biological Chemistry, University of California Los Angeles, CA, USA) made to the catalytic fragment of rat HMGCR and cross-reacted with human HMGCR. Secondary antibodies for immunofluorescence microscopy: Cy2- and Cy3-labeled donkey α-rabbit and α-mouse IgG, respectively (Jackson ImmunoResearch, Philadelphia, PA, USA). For Western blot analysis, horseradish peroxidase (HRP)-conjugated secondary antibodies (Dako, Glostrup, Denmark) were used. 

### 2.4. Western Blot Analysis

After treatments, cells were washed with PBS and lysed in RIPA buffer, which was supplemented with a protease inhibitor cocktail (Complete Protease Inhibitor Cocktail Tablets, Roche, Basel, Switzerland), 100 µM Leupeptin (Roche), 2 mM freshly prepared PMSF (phenylmethylsulfonyl fluoride, Sigma) and 10 mM DTT (dithiothreitol, Boehringer Ingelheim, Ingelheim am Rhein, Germany) for 15 min on ice. The lysate was transferred to a 1.5 mL syringe, followed by passage through a 27.5 gauge needle 3 times to shear the genomic DNA mechanically. Then, the lysate was centrifuged for 10 min at full speed at 4 °C. The supernatant was collected and the protein concentration was determined using Bradford protein assay. For SDS polyacrylamide gel electrophoresis (PAGE), the samples were incubated with Laemmli sample buffer at 37 °C for 30 min. Protein lysates were separated into polyacrylamide gels at an appropriate percentage and transferred to nitrocellulose membranes (0.2 micron, GE healthcare, Little Chalfont, PA, USA), blocked with 4% skim milk in TBS-T (TRIS-buffered saline with 0.05% *w*/*v* Tween 20) for 1 h, followed by incubation with primary antibodies overnight at 4 °C. Membranes were incubated with the corresponding HRP-conjugated secondary antibodies (Dako) for 2 h at room temperature. Millipore HRP substrate peroxide solution was used as a chemiluminescence-enhancing reagent. Relevant protein bands were detected and visualized on a Chemidoc XRS+ detection system (Bio-Rad, Hercules, CA, USA) and analyzed with Image Lab 5.1 analysis software (Bio-Rad). 

### 2.5. Density Gradient Centrifugation for Organelle Fractionation

Density gradient centrifugations were carried out as described earlier [78]. In brief, OptiPrep (Axis-Shield) density gradient solutions were prepared based on the same buffer composition as the homogenization buffer (HB; 250 mM sucrose, 1 mM EDTA (free acid), 0.1% ethanol, 0.1 mM PMSF, complete protease inhibitor mixture (Roche Applied Science, catalog number 04693132001), 3 mM imidazole and KOH to pH 7.4) and further diluted with HB. Two milliliters of postmitochondrial supernatant (containing 4 mg of protein) was layered on top of a prebuilt 10–30% isoosmotic OptiPrep gradient, and a 40% OptiPrep cushion was used. Centrifugation was carried out at 25,000 rpm for 90 min in a Beckman VTi 65.2 vertical rotor (without braking below 3000 rpm). Gradient fractions were collected from the bottom, and the respective organellar markers were identified by Western blot analysis.

### 2.6. PTS2 Reporter Constructs and Analysis of Peroxisomal Targeting in COS-7 Cells 

A PTS2 reporter construct was generated previously [79], in which the first 30 AAs of rat thiolase B were cloned in front of EGFP, and the PTS2 nonapeptide was flanked by two restriction sites (*PstI* and *EcoRI*) allowing the exchange of nonapeptides. The plasmid encoding the reporter protein with the human thiolase PTS2 (RLQVVLGHL) was used as a positive control for peroxisomal localization of EGFP upon transfection into COS-7 cells [79]. The oligonucleotide sequences encoding the HMGCR-PTS2A (RATFVVGNS) and the HMGCR-PTS2 B (RGVSIRRQL) as well as the mutated HMGCR-PTS2AMut (DDIFVVGNS) including the restriction enzyme sequence for *PstI* and *EcoRI* were cloned into the reporter construct. The reporter plasmids were transfected into COS-7 cells and the colocalization with the peroxisomal membrane protein PMP70 was investigated by immunofluorescence microscopy.

### 2.7. Statistical Analysis

Statistical computations were conducted using SPSS Statistics version 20 (IBM Corporation, Armonk, NY, USA). Data sets were tested for normal distribution by Levene’s test first. One-way analysis of variance (ANOVA) with LSD (least significant difference) post hoc comparison was performed for normally distributed data to compare mean values among all measured variables. A non-parameter Mann–Whitney U test was used for multiple comparisons among the different groups if the data sets did not have similar variances. Quantitative data and graphical results are represented as mean ± standard error of mean (s.e.m.) or mean ± standard deviation (s.d.). Sample size is indicated in the figure caption. Statistically significant results are indicated as * *p* < 0.05, ** *p* < 0.01, *** *p* < 0.001. 

## 3. Results

### 3.1. HMG-CoA Reductase Is Bilocalized to ER and Peroxisomes in Human Differentiated THP-1 Cells under Conditions of Low Cholesterol and Lovastatin Treatment

As cholesterol metabolism is of particular importance for macrophages, we first investigated the subcellular localization of HMGCR in THP-1 monocytic cells differentiated with PMA to macrophage-like cells. Differentiated THP-1 cells were incubated in lipid-depleted medium for 7 days and the HMGCR inhibitor lovastatin was added for the last 16 h. Under these conditions, HMGCR is known to be highly abundant due to its high expression and low degradation rates. Using an antibody against HMGCR obtained from Edwards (anti-HMGCR-E) and one for the ER marker protein PDI, we observed the expected localization of HMGCR in the ER (Figure 1a–c). However, additional punctuated staining was observed that did not co-localize with PDI. When co-staining the cells with the antibody against HMGCR and one directed against the peroxisomal membrane protein ABCD1, the punctate staining co-localized with ABCD1 (Figure 1d–f). This demonstrates peroxisomal HMGCR’s immunoreactivity and suggests a dual localization of HMGCR to the ER and peroxisomes under conditions of cholesterol depletion and HMGCR inhibition. To corroborate these results and to rule out an unspecific peroxisomal labeling by the polyclonal HMGCR antibody, we additionally used a monoclonal anti-HMGCR antibody (anti-HMGCR-A9) directed against the C-terminal domain. First, we showed the predominant ER localization by using co-staining with the ER marker calnexin (Appendix A). We next performed co-staining with an anti-PMP70 antibody labeling another peroxisomal membrane protein (Figure 1g–i). Again, HMGCR’s immunoreactivity was consistently found to be colocalized with the peroxisomal marker. Colocalization of PMP70 with the ER marker protein PDI verified that the staining of the two organelles is specific (Appendix A). Finally, these findings were confirmed by confocal microscopic imaging (Figure 1j,k). Of note, in each staining, approximately 70% of cells showed dual HMGCR localization, whereas in about 30% an exclusive ER localization of the enzyme was observed. Altogether, these results demonstrate a clear dual localization of HMGCR under low-cholesterol conditions with lovastatin treatment. 

### 3.2. The Fraction of Peroxisomal HMGCR Increases with the Differentiation State of THP-1 Cells

It is known that PMA-induced differentiation toward macrophages increases HMGCR activity and cholesterol synthesis [80]. Thus, we next set out to quantify the percentage of cells demonstrating peroxisomal HMGCR localization under low-cholesterol conditions and statin treatment and in relation to PMA differentiation. To this end, we quantified the number of cells with peroxisomal HMGCR in undifferentiated THP-1 cells cultured for 3 days in lipid-depleted medium and for the last 16 h with lovastatin, and compared it to PMA-differentiated THP-1 cells cultured for 3 or 7 days in lipid-depleted medium with 16 h lovastatin treatment. We found that 14% of the undifferentiated THP-1 cells presented peroxisomal HMGCR localization (Figure 2a–c,j). This number increased with differentiation to 50% after 3 days and 68% after 7 days (Figure 2d–j). These findings indicate that the amount of peroxisomal HMGCR reaching the detection threshold depends on the metabolic state or differentiation state of the THP-1 cells or both. The heterogeneity in the differentiation process possibly explains why not all THP-1 cells show peroxisomal HMGCR. 

### 3.3. A Truncated HMGCR Is Located to Peroxisomes

To verify this result of a dual localization of HMGCR under low-cholesterol conditions with lovastatin treatment in differentiated THP-1 cells with a second independent method, we performed density gradient centrifugation for organellar separation as described previously [78]. Fractions of the density gradient were loaded onto a polyacrylamide gel and Western blot analysis was performed. Fractions containing ER were identified by an anti-GRP78 antibody (ER chaperone, 78-kDa glucose regulated protein), peroxisomal fractions by the anti-PMP70 antibody and mitochondria fractions by an anti-ATP-synthase antibody (Figure 3). Full-length HMGCR and truncated versions thereof were detected by Edwards’ anti-HMGCR antibody. Whereas the full-length 96 kDa HMGCR was only visible in the ER-containing fractions, another band with a molecular weight of 76 kDa was observed in the peroxisomal fractions (Figure 3 and Appendix A) and probably reflects a truncated version of HMGCR. This suggests that the HMGCR immunoreactivity in peroxisomes is caused by a truncated and not full-length version of the protein.

### 3.4. The Percentage of Cells with Detectable Peroxisomal HMGCR under Low-Cholesterol Conditions and Lovastatin Treatment Is Cell-Type-Specific

To further investigate whether the peroxisomal HMGCR localization is a general phenomenon observed under conditions of low cholesterol and statin treatment, we investigated differentiated human monocyte-derived U937 cells, human microglia-like CHME-3 cells, the hepatoma cell line HepG2 as well as HeLa and HEK-293 cells, all after 3 days of cultivation in lipid-depleted media and 16 h of lovastatin treatment. Whereas in U937, HeLa and HEK-293 no co-localization of HMGCR with peroxisomes was detectable, the enzyme was bilocalized to both ER and peroxisomes in a small percentage of CHME-3 (5%) and HepG2 (3%) cells (Table 1). To verify that the bilocalization of HMGCR is not only found in human cell lines but is also observed in primary human cells, we isolated monocytes from leukocyte reduction chambers of human healthy donors and differentiated the cells for 7 days using M-CSF to mature macrophages before culturing them in lipid-depleted medium for 3 days and finally adding lovastatin for 8 h. We found that in 65% of monocyte-derived primary macrophages, HMGCR was found in peroxisomes (Figure 4a–d, Table 1). This percentage of cells with detectable peroxisomal HMGCR is comparable to that observed in differentiated THP-1 cells (68%; Table 1). In primary human fibroblasts after 3 days of culturing in lipid-depleted medium and 16 h of lovastatin treatment, we observed a peroxisomal localization of HMGCR in 33% of cells (Table 1). Taken together, differentiated human macrophages and the macrophage-like THP-1 cells, but not U937 cells, along with primary human fibroblasts exhibited the highest fraction of cells with a detectable peroxisomal HMGCR localization.

### 3.5. PEX7 Is Required for the Peroxisomal Localization of Human HMGCR

Next, we addressed the molecular mechanism of peroxisomal import. We supposed that the fragment corresponds to the soluble domain of HMGCR without transmembrane domains, and thus a peroxisomal targeting signal (PTS) at the C-terminus (type 1, PTS1) or close to the newly generated N-terminus (type-2, PTS2) was expected. As the extreme C-terminus of human HMGCR does not resemble a PTS1, we expected a PTS2 and tested this hypothesis by studying primary human fibroblasts from a patient suffering from type-1 rhizomelic chondrodysplasia punctata (RCDP1), in which the PTS2 receptor PEX7 is mutated. Indeed, in contrast to control primary fibroblasts with peroxisomal HMGCR localization (Figure 5a–c), the fibroblasts from the RCDP1 patient did not show peroxisomal HMGCR (Figure 5d–f). As in PEX7-deficient cells the peroxisomal part of the ether phospholipid synthesis is impaired, leading to plasmalogen deficiency, we excluded the possibility that this causes the absence of peroxisomal HMGCR. Thus, we next investigated primary human fibroblasts from a patient with RCDP-2 (rhizomelic chondrodysplasia punctata type-2) harboring mutations in the enzyme glyceronephosphate O-acyltransferase (GNPAT) required for ether phospholipid synthesis via a PTS1/Pex5-dependent mechanism. In these cells, HMGCR was also detected in peroxisomes (Figure 5g–i). These findings demonstrate that the peroxisomal localization of the truncated HMGCR under conditions of low cholesterol specifically depends on PEX7.

### 3.6. The C-Terminal Domain of HMGCR Contains a Functional PTS2 Close to Its N-Terminus

As peroxisomal import of the truncated HMGCR depends on PEX7, a functional PTS2 is expected within the truncated HMGCR. Two putative PTS2 sequences were identified in the linker region and at the beginning of the catalytic domain of human HMGCR, (Figure 6a) which we termed HMGCR-PTS2-A (RATFVVGNS, 412–420 AAs) and HMGCR-PTS2 B (RGVSIRRQL, 490–498 AAs), respectively (Figure 6b). To investigate the functionality of these PTS2-like motifs, we used a reporter protein consisting of the first 30 AAs of rat thiolase in front of EGFP72 (Figure 6b). When the reporter protein harboring HMGCR-PTS2-A was expressed in COS-7 cells, EGFP showed a punctate staining pattern, which co-localized with PMP70 (Figure 6c–e). This indicated that the nonapeptide sequence can mediate peroxisomal import of the reporter protein and thus can be considered as a functional PTS2. In contrast, when the reporter protein harbored HMGCR-PTS2-B, the reporter protein was found in the cytosol and nucleus (Figure 6f–h), indicating that the nonapeptide is not a functional PTS2. The introduction of three point mutations in the HMGCR-PTS2-A sequence (HMGCR-mut-PTS2-A, DDIFVVGNS) caused failure in targeting the reporter protein into peroxisomes (Figure 6i–k). 

These findings demonstrated that an effective PTS2 is encoded in the linker region of human HMGCR, connecting the membrane-embedded N-terminal part of HMGCR and the catalytic C-terminal domain. Irrespective of which processing site in HMGCR is critical under these conditions, the PTS2 motif is located in close proximity to the novel N-terminal end of the soluble HMGCR domain and the location within a flexible linker exposes the PTS2 to the receptor PEX7. This should allow for the targeting of the truncated soluble HMGCR containing the complete catalytic domain into peroxisomes.

## 4. Discussion

HMGCR has been detected in peroxisomes by using electron microscopy and immunogold labelling. The cellular metabolic conditions promoting bilocalization and the mechanism leading to a peroxisomal transport, however, are unknown. There is no evidence for a second *HMGCR* gene in mammals [81]. Here, we demonstrate that under conditions of low cholesterol and statin treatment, a truncated form of HMGCR is targeted to peroxisomes. This finding is observed by co-immunofluorescence microscopy using two different antibodies recognizing HMGCR and by density gradient centrifugation. The peroxisomal HMGCR is not the full-length protein but a truncated fragment of approximately 76 kDa that was found in peroxisome-enriched fractions containing PMP70. In contrast, the full-length HMGCR protein of 96 kDa was exclusively present in fractions containing GRP78, a well-established ER protein. Unfortunately, we were not able to identify the N-terminus of the truncated peroxisomal HMGCR variant. Based on the apparent molecular weight, the truncated 76 kDa HMGCR that we observed in our experiments is different from the 90 kDa peroxisomal HMGCR variant observed in UT2* cells or in CHO cells by Krisans’ group [82]. Importantly, we only observed the peroxisomal localization of the truncated soluble HMGCR under low cholesterol in the presence of lovastatin, a condition known to increase production and to stabilize HMGCR, as the sterol depletion prevents HMGCR from binding to INSIG proteins and the associated E3 ubiquitin ligases (gp78, TRC8 and RNF145) [83]. Thus, under these specific conditions, the standard degradation pathway of HMGCR is not expected to generate such truncated peroxisomal HMGCR. However, it is tempting to speculate that under these conditions residual processing continues, but a low degradation rate of the HMGCR fragment causes its accumulation in the cytosol. 

We demonstrate that the peroxisomal targeting of HMGCR is PEX7-dependent. PEX7 is the soluble receptor, which binds peroxisomal matrix proteins harboring a PTS2 motif close to the N-terminus [84]. We identified a functional PTS2 in the flexible linker region of HMGCR close to the putative new N-terminus of the truncated HMGCR. This PTS2 motif is able to target a reporter protein into peroxisomes and is embedded within a flexible domain, which is a property amply found in typical PTS2-carrying proteins [79]. The exact N-terminus of the truncated HMGCR has not yet been identified, and the suggested PTS2 (RATFVVGNS) deviates from the traditional PTS2 consensus sequence ([R/K]-[L/V/I/Q]-X-X-[L/V/I/H/Q]-[L/S/G/A/K]-X-[H/Q]-[L/A/F]) [84,85]. Thus, the contribution of another targeting sequence cannot be excluded. Based on the apparent molecular weight and the epitope recognized by the monoclonal HMGCR-A9 antibody, the HMGCR fragment should contain the catalytic domain. The peroxisomal polypeptide has a higher molecular weight than the truncated 62 kDa fragment that is released from the ER under high-cholesterol conditions [86]. Importantly, this 62 kDa fragment has been shown to be catalytically active even when it is further processed to a 53 kDa HMGCR C-terminal fragment [86]. 

We have previously demonstrated that cholesterol can be synthesized from radioactively labelled acetate generated by peroxisomal β-oxidation of the labelled very-long-chain fatty acid C24:0. Of note, the amount of cholesterol containing acetyl units generated in peroxisomes was not affected by lovastatin treatment, whereas the amount of cholesterol containing mitochondrially generated acetyl-CoA or acetyl-CoA derived from exogenously added acetate was strongly diminished under these conditions [65]. Unfortunately, the metabolic embedding of peroxisomal HMGCR is unclear. It seems reasonable to assume that the structural similarity between 3-hydroxy-3-methylglutaryl-CoA and branched-chain fatty acids or the side chain of the bile acid precursors di- or trihydroxycholestanoic acid might allow import via the same peroxisomal ABC transporter, ABCD3/PMP70 [87], to gain access to peroxisomal HMGCR. Under normal conditions, the product of HMGCR, mevalonate, is further processed by the enzyme mevalonate kinase in the next step of isoprenoid biosynthesis. Although some publications have suggested a peroxisomal localization of this protein [49,88], we share the view that these studies do not provide sufficient evidence for a peroxisomal localization [57]. We rather suggest that mevalonate is transferred to the ER and that this transfer is supported by the tethering and physical interaction between peroxisomes and ER [70]. In this way, mevalonate could reach the other enzymes at the ER which are not tightly regulated. Thus, under conditions of low cholesterol and statin treatment, the truncated peroxisomal HMGCR might escape statin inhibition and contribute to isoprenoid production. As the appearance of peroxisomal HMGCR only occurs under very specific conditions when particularly large amounts of full-length HMGCR are localized at the ER, it could also be a non-functional bystander effect of an alternative turnover process under conditions of cholesterol depletion and statin binding. Also, alterations in the ER membrane and ER stress and an overloaded transport system might contribute to the findings. The open question of the physiologic significance is a limitation of this study, as we were not able to resolve this issue.

In this study, we contribute to the long-standing question of whether or not HMGCR can be localized to peroxisomes in addition to the ER. We identified cell types and a condition where this reliably occurs. These findings add another puzzle piece to the complex regulation of intracellular cholesterol homeostasis. Isoprenoid biosynthesis may be another example, next to ether phospholipid biosynthesis [42], that requires tight interactions between peroxisomes and the ER for effective transfer of certain metabolites. These findings under low-cholesterol conditions and statin treatment might be relevant for human patients with hypercholesterolemia receiving statins in combination with a low-cholesterol diet and/or the antihyperlipidemic drug colestipol. 

## Figures and Tables

**Figure 1 biomolecules-14-00244-f001:**
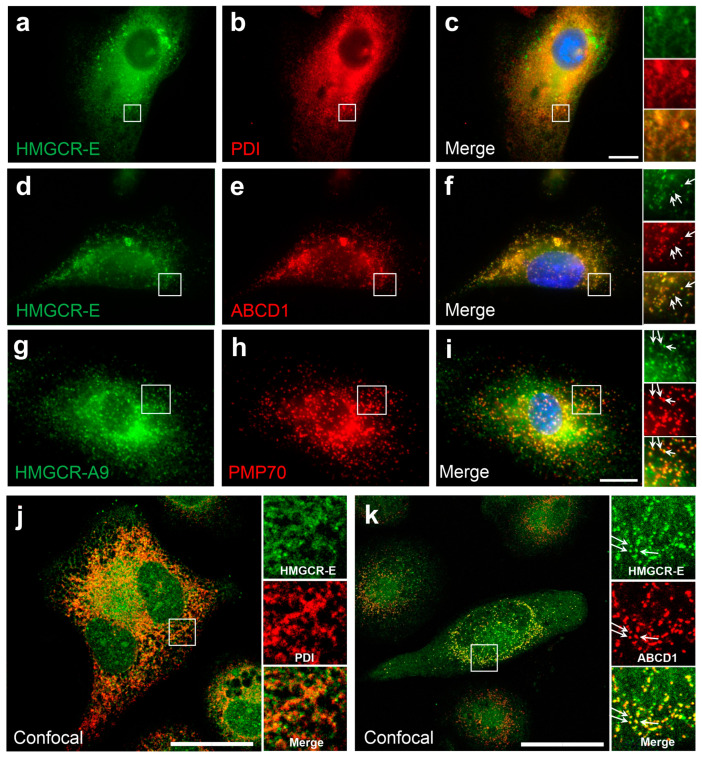
Localization of human HMG-CoA reductase in peroxisomes of differentiated THP-1 cells: (**a**–**j**) Cells of the human monocytic cell-line THP-1 were differentiated using 40 ng/mL PMA and cultivated in lipid-depleted medium for 7 days with 5 µM lovastatin added for last 16 h to increase endogenous HMG-CoA reductase (HMGCR). The subcellular distribution of HMGCR was investigated by immunofluorescence microscopy (**a**–**i**) and confocal microscopy (**j,k**) using different antibody pairs: (**a**–**c**) rabbit polyclonal α-HMGCR-E (green) with mouse α-PDI (red) labeling the ER; (**d**–**f**) α-HMGCR-E (green) with mouse α-ABCD1 (red) labeling peroxisomes; (**g**–**i**) murine monoclonal α-HMGCR-A9 (green) with rabbit polyclonal α-PMP70 (red) labeling peroxisomes; (**j**) α-HMGCR-E (green) and α-PDI (red); and (**k**) α-HMGCR-E (green) and α-ABCD1 (red). The co-staining demonstrates that under these conditions HMGCR is co-localized with both an ER marker protein (**a**–**c**,**j**) and two different peroxisomal marker proteins (**d**–**f**,**g**–**i,k**). White arrows: punctate peroxisomal staining pattern. Scale bars: 20 μm.

**Figure 2 biomolecules-14-00244-f002:**
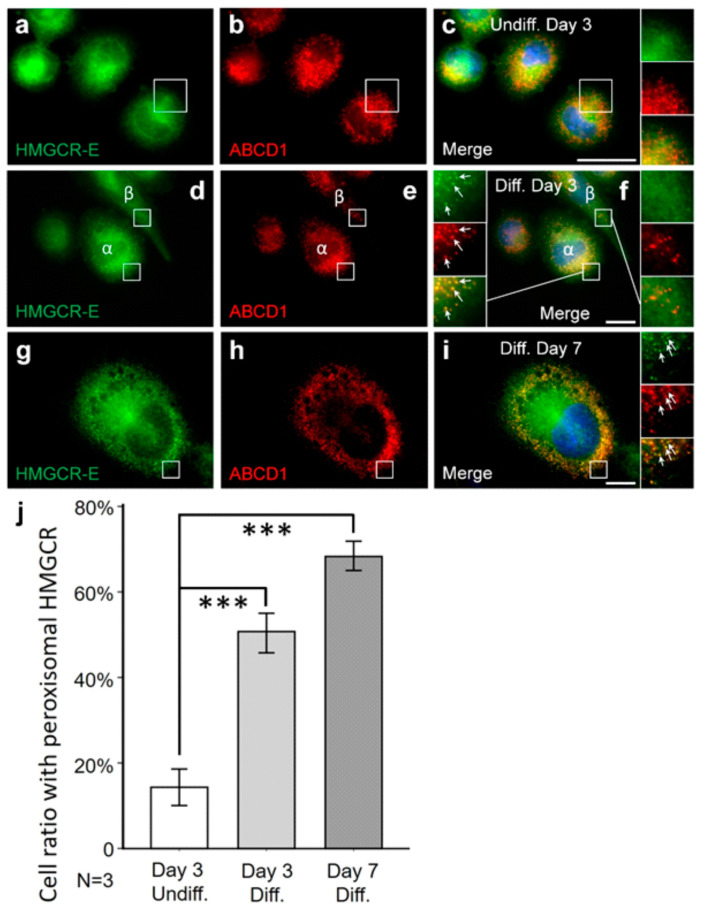
The fraction of THP-1 cells with peroxisomal HMGCR increases with PMA-mediated differentiation. THP-1 cells were incubated in lipid-depleted medium (LDM) for 3 days followed by 16 h with lovastatin (**a**–**c**), differentiated for 3 days with PMA and then incubated in LDM for 3 days with 16 h lovastatin treatment (**d**–**f**) or differentiated for 7 days with PMA and then incubated in LDM for 7 days with 16 h lovastatin (**g**–**i**). Undifferentiated THP-1 cells were smaller in size (compare **c**,**f**,**i**) and also presented lower percentage of cells with peroxisomal HMGCR (α-HMGCR-E (green) and α-ABCD1 (red) antibodies). Manual counting verified a significantly higher fraction of differentiated cells with peroxisomal α-HMGCR signal (**j**). (**f**) After 3 days of differentiation, THP-1 cells with (**f**-**α**) and without (**f**-**β**) peroxisomal labelling by the α-HMGCR antibody were present on the same slide. White arrows: punctate peroxisomal staining pattern. Scale bars: 20 μm. (**j**) For each condition, 3 individual coverslips were investigated, 20 areas were randomly selected and at least 36 cells were evaluated per coverslip. Cells were identified by DAPI staining and the fraction of cells with peroxisomal HMGCR immunoreactivity (α-HMGCR-E (green) and α-ABCD1 (red) antibodies) was estimated by manual counting. Comparison between these three groups revealed an increasing percentage of THP-1 cells with peroxisomal HMGCR (from undifferentiated 14 ± 9.5%, 3 days diff. 50 ± 10.9%, 7 days diff. 68 ± 9.5%) after long-term differentiation under low-cholesterol conditions. Error bars indicate SEM; *** *p* < 0.001.

**Figure 3 biomolecules-14-00244-f003:**
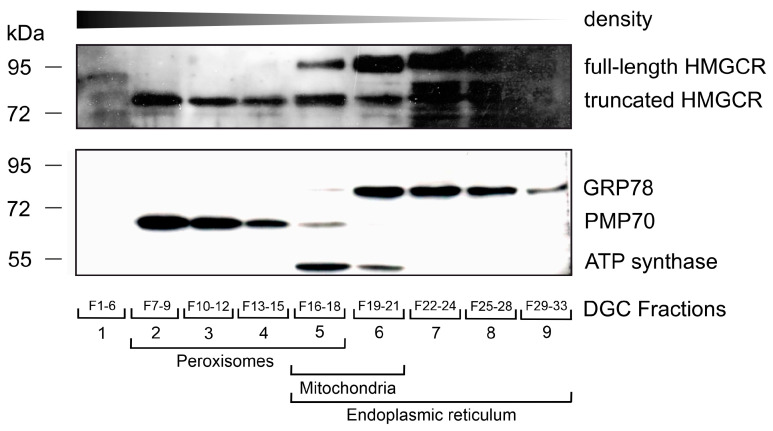
In differentiated THP-1 cells grown under low-cholesterol conditions and lovastatin treatment, a truncated form of HMGCR was found in peroxisome-enriched fractions obtained by density gradient centrifugation (DGC). THP-1 cells were differentiated and incubated in lipid-depleted medium for 7 days. An amount of 5 µM lovastatin was provided for the last 16 h before harvesting (low-cholesterol conditions). DGC was used to enrich organelles and separate fractions with different compositions according to their density. Fractions were then pooled and Western blots were performed using α-GRP78, α-PMP70 and α-ATP synthase antibodies to identify fractions enriched in ER (fraction 16–33, lane 5–9), peroxisomes (F7–18, lane 2–5) and mitochondria (F16–21, lane 5–6), respectively. Whereas full-length HMGCR with a size of about 97 kDa was found in fractions enriched in the ER (Lanes 5–9), a truncated form of HMGCR with a size of about 76 kDa was additionally found in peroxisome-enriched fractions (Lane 2–5). Western blot original images can be found in Appendix A. The sizes of the molecular weight marker are indicated on the left side.

**Figure 4 biomolecules-14-00244-f004:**
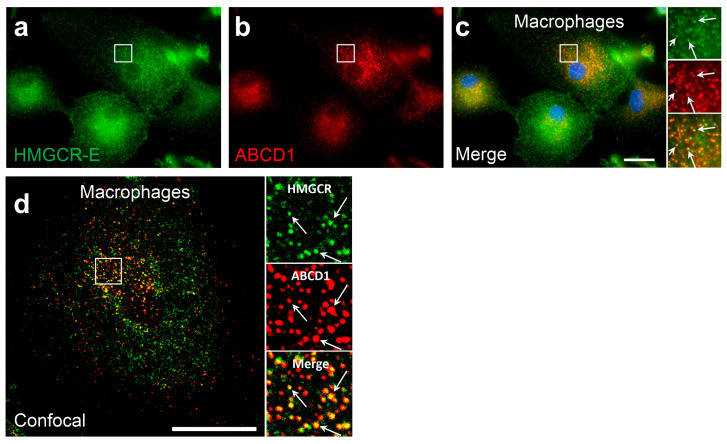
In in vitro-differentiated human macrophages, HMGCR also co-localizes with peroxisomes: (**a**–**d**) Immunofluorescence microscopic analysis of primary monocyte-derived macrophages using α-HMGCR-E (green) and α-ABCD1 (red) antibodies: Primary human CD14+ monocytes were differentiated to macrophages using RPMI supplemented with 50 ng/mL macrophage colony-stimulating factor (M-CSF) for 7 days, then incubated for 3 days with lipid-depleted medium followed by an incubation for 8 h with 5 µM lovastatin. In epifluorescence (**a**–**c**) and confocal (**d**) microscopic analyses, α-HMGCR immunoreactivity colocalized with the peroxisomal marker ABCD1 (α-HMGCR-E (green) and α-ABCD1 (red) antibodies). White arrows: punctate peroxisomal staining pattern. Scale bars: 20 μm.

**Figure 5 biomolecules-14-00244-f005:**
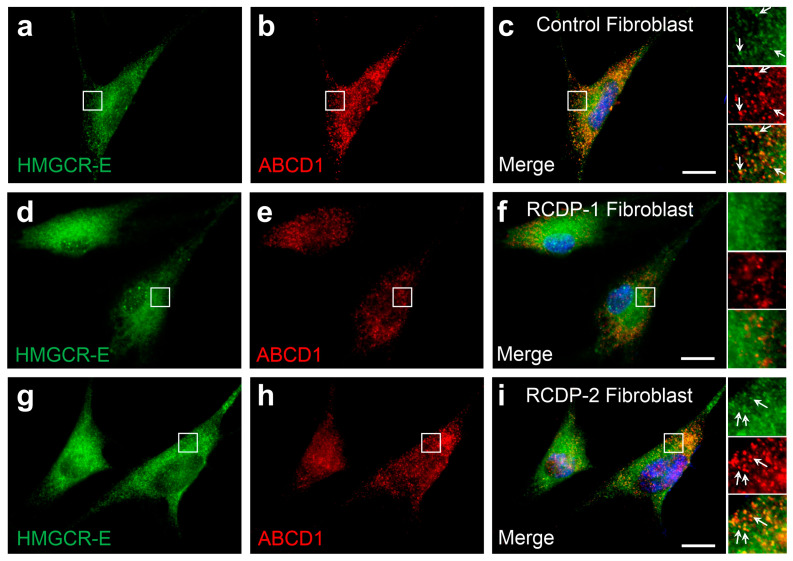
The peroxisomal localization of HMGCR requires the PTS2 receptor PEX7. Immunofluorescence microscopic analysis of primary human dermal fibroblasts using α-HMGCR-E (green) and α-ABCD1 (red) antibodies: (**a**–**c**) Human fibroblasts obtained from a healthy subject were treated with lipid-depleted medium for 3 days and 5 µM lovastatin was added for 16 h. A fraction of cells with α-HMGCR immunoreactivity showed a punctate staining pattern that co-localized with the peroxisomal marker protein ABCD1. (**d**–**f**) However, in PEX7-deficient human fibroblasts obtained from a patient suffering from rhizomelic chondrodysplasia punctata type-1 (RCDP-1) grown under the same conditions, α-HMGCR immunoreactivity showed a diffusely distributed pattern but no co-localization with the peroxisomal marker protein in all cells. (**g**–**i**) In contrast, in GNPAT-deficient fibroblasts obtained from a patient suffering from RCDP-2 and grown under the same conditions, α-HMGCR immunoreactivity was also co-localized with peroxisomes in a fraction of cells. White arrows: punctate peroxisomal staining pattern. Scale bars: 20 μm.

**Figure 6 biomolecules-14-00244-f006:**
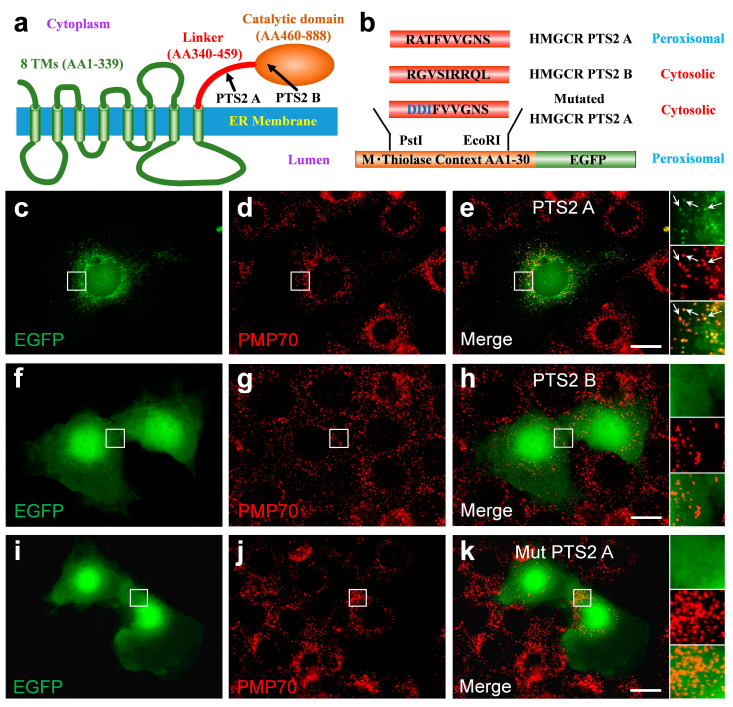
A functional type-2 peroxisome-targeting signal (PTS2) motif is encoded in the linker domain of human HMGCR: (**a**) Domain structure of human HMGCR consisting of a highly conserved N-terminal domain comprising eight transmembrane spans, a flexible linker region and a conserved catalytic domain; two putative PTS2 were identified, one in the linker region (HMGCR-PTS2-A: 411–420 AAs) and one at the beginning of the catalytic domain (HMGCR-PTS2-B: 490–498 AAs). (**b**) Schematic representation of the PTS2 reporter protein [79] consisting of the first 30 AAs of rat thiolase in front of EGFP. To test the functionality of putative PTS2 motifs, the candidate nonapeptides were inserted into the reporter protein to substitute the endogenous PTS2. (**c**–**k**) Immunofluorescence microscopic analysis of the subcellular distribution of various reporter proteins, which encode HMGCR-PTS2-A (RATFVVGNS) (**c**–**e**), HMGCR-PTS2-B (RGVSIRRQL) (**f**–**h**) or a variant of HMGCR-PTS2-A, in which two negative charges were introduced to obtain the inactive HMGCR-mutated-PTS2-A (DDIFVVGNS); COS-7 cells were transfected with expression plasmids for these reporter protein variants encoding the PTS2-like motifs and the distribution of EGFP was investigated. A punctate staining pattern that co-localized with the peroxisomal marker protein PMP70 was only found for the reporter protein encoding HMGCR-PTS2-A. When the reporter protein encoded the HMGCR-PTS2-B or HMGCR-mutated-PTS2-A, it was found evenly distributed across the cytosol and the nucleus but did not co-localize with PMP70. White arrow: punctate peroxisomal staining pattern. Scale bar: 10 µm.

**Table 1 biomolecules-14-00244-t001:** Percentage of peroxisomal HMGCR in different human cell lines under certain conditions.

Cell Line	Cell Type	Differentiation/Stimulation	LDM Treatment	Lovastatin Treatment	Perc. Peroxisomal HMGCR (±STDEV)	Quantification
	Monocyte	7 d differentiation	7 days	16 h	68 ± 8.7%	147 (*n* = 3)
THP-1		3 d differentiation	3 days	16 h	50 ± 10.9%	211 (*n* = 3)
		No differentiation	3 days	16 h	14 ± 9.5%	206 (*n* = 3)
U937	Monocytic cell	3 d differentiation	3 days	16 h	0%	ND
HeLa	Epithelial cell	No differentiation	3 days	16 h	0%	ND
HEK-293	Embryonic kidney	No differentiation	3 days	16 h	0%	ND
CHME-3	Microglia cell	1 d stimulation	3 days	16 h	<5%	ND
HepG2	Liver cell	No differentiation	3 days	16 h	<3%	ND
	Primary macrophage	7 d differentiation	3 days	8 h *	65 ± 8.8%	320 (*n* = 3)
	Primary control fibroblast	No differentiation	3 days	16 h	33 ± 10.0%	201 (*n* = 3)

* For primary human in vitro-differentiated macrophages, only 8 h lovastatin incubation was carried out because of the sensitivity of this primary cell type.

## Data Availability

All data presented in the manuscript and/or Appendix A are available upon request.

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
