# Peer review of "Peroxisomal Localization of a Truncated HMG-CoA Reductase under Low Cholesterol Conditions"

_biomolecules, 2024, doi:10.3390/biom14020244_

Round 1

Reviewer 1 Report

Comments and Suggestions for Authors

The manuscript of Wang et al. details the appearance of HMGCoA reductase (HMGCR) in peroxisomes.  The authors have confirmed previous observations and have found conditions where they can reproducibly observe HMGCR. The authors have provided solid evidence to support their conclusions. The manuscript is well-written with no major issues.  I found the observation intriguing as HMGCR is the major regulatory step of the cholesterol biosynthesis pathway and this could have serious consequences upon statin use.  However, I have concerns that the presence of HMGCR was only observed under the unphysiological condition of cholesterol deprivation in the presence of a statin.  My comments are below.

1. In the presence of statin, HMGCR protein production is upregulated significantly in an effort to produce more cholesterol.  HMGCR is proteolytically degraded upon high cholesterol levels, but under cholesterol deficient conditions, HMGCR accumulates.  It has been shown that ER stress occurs because of the overloading of ER membranes with HMGCR protein.  It is important that the authors show exactly how much more HMGCR protein there is in their cells used in these experiments treated with statin under cholesterol deprivation (compared to appropriate controls) by western blot.  If there is 20 fold more HMGCR protein in the ER membrane, it may be a simple overloading of transport systems that lead to HMGCR ending up in the peroxisomes. 

2. Does the truncated HMGCR have activity (ie. makes mevalonate) in the peroxisomes? This seems like a critical question determining whether this HMGCR is active and therefore physiologically relevant, or is simply a truncated artifact on its way to being disposed.

3. HMGCR is sometimes found in the lipid droplets (or an associated fraction) on its way to being proteolytically degraded (Hartman et al. DOI 10.1074/jbc.M110.134213).  Is there a way for the authors to measure ERAD of HMGCR, activity of VCP/p97, or something similar to show whether this pathway is upregulated?

4. Is expression of Insig1 or Insig2 upregulated in response to statin use and cholesterol deprivation?

5. The cells are chronically deprived of cholesterol. Is there any evidence that the cells are sick, apoptotic, or have impaired normal signaling functions due to the lack of cholesterol?  Is the displacement of HMGCR due to the lack of cholesterol or to an over expression of HMGCR?

6. Could the authors comment if the condition of the cells (statin use and cholesterol deprivation) corresponds to any condition in vivo?

Reviewer 2 Report

Comments and Suggestions for Authors

The manuscript by Wang et al. thoroughly investigates the localization of HMGCR under conditions of low cholesterol with statins. The peroxisomal location has been observed previously but not conclusively determined, so this study was warranted to fully investigate this question. Interestingly, not all cell types respond with to low cholesterol conditions by directing HMGCR to the peroxisome, which could be helpful in investigating the functions of the peroxisomal HMGCR, which are still undetermined. Overall, the authors have done an in-depth study of the peroxisomal targeting of HMGCR, including the PTS2-dependent Pex7 import into the peroxisome. Unfortunately, the piece of data showing the possible truncation of peroxisomal HMGCR is a western blot that is difficult to interpret, and if possible, should be redone, as this is an important part of the targeting to the peroxisome. Overall, this is a nice piece of work that can inspire the future investigation of the function of the peroxisomal HMGCR. 

Comments on the Quality of English Language

A couple minor typos were noted. Overall, it was very easy to read and understand. 

Reviewer 3 Report

Comments and Suggestions for Authors

A key regulatory step in the biosynthesis of cholesterol is mediated by the endoplasmic reticulum (ER) enzyme, HMG-CoA reductase (HMGCR). A recurrent theme in the literature is that peroxisomes might also participate in sterol biosynthesis. Wang et al. have investigated this issue with an immunocytochemical analysis of the disposition of the enzyme and its catalytic C-terminal fragment in various mammalian cells. They detect no significant amount of enzyme or the fragment in unperturbed cells. However, they can promote the physiologically-targeted appearance of the latter (but not the holo- enzyme) in peroxisomes after stimulating its overproduction by starving cells of the sterol plus blocking HMGCR with a statin.  The project is expert: well conceived and well executed. Their cytochemical and subcellular fractionation techniques are impressive and convincing. Furthermore, Wang et al. deal with the literature effectively. Importantly, they are careful and conservative in their interpretation of their data and its significance. This is key because of the relationship of their data to the ongoing question of peroxisomal sterol biogenesis, mentioned above.

The significance of the study can be divided into two parts. The positive aspect is as described above: an excellent analysis of a highly-constructed cell state. While the authors may not have sought this outcome, their convincing inability to detect either the holo-enzyme or its fragment  in peroxisomes will be important to the field. The negative aspect is that HMGCR is not detectable in peroxisomes in unperturbed cells and even in those overexpressing the enzyme. Thus, the appearance of the catalytic fragment in the peroxisomes of overexpressing cells, while shown to be physiologically-mediated, is of unknown physiologic significance. Wang et al. say as much even though they tried to put the best possible face on their results. All in all, this study will be of interest and significance to those in the field.

Minor points : Line 55 it should be "...in peroxisomes"   There is a typo in line 427.  recongnizing.  
